# Mapping the Photochemistry of European Mid-Latitudes Rivers: An Assessment of Their Ability to Photodegrade Contaminants

**DOI:** 10.3390/molecules25020424

**Published:** 2020-01-20

**Authors:** Luca Carena, Davide Vione

**Affiliations:** Dipartimento di Chimica, Università di Torino, Via Pietro Giuria 5, 10125 Torino, Italy; luca.carena@unito.it

**Keywords:** Indirect photochemistry, hydroxyl radicals, direct photolysis, photochemical modeling, dissolved organic matter, CDOM

## Abstract

The abiotic photochemical reactions that take place naturally in sunlit surface waters can degrade many contaminants that pose concern to water bodies for their potentially toxic and long-term effects. This works aims at assessing the ability of European rivers to photoproduce reactive transient intermediates, such as HO^•^ radicals and the excited triplet states of chromophoric dissolved organic matter (^3^CDOM*), involved in pollutant degradation. A photochemical mapping of the steady-state concentrations of these transients was carried out by means of a suitable modeling tool, in the latitude belt between 40 and 50°N. Such a map allowed for the prediction of the photochemical lifetimes of the phenylurea herbicide isoproturon (mostly undergoing photodegradation upon reaction with HO^•^ and especially ^3^CDOM*) across different European countries. For some rivers, a more extensive dataset was available spanning the years 1990–2002, which allowed for the computation of the steady-state concentration of the carbonate radicals (CO_3_^•^^−^). With these data, it was possible to assess the time trends of the photochemical half-lives of further contaminants (atrazine, ibuprofen, carbamazepine, and clofibric acid). The calculated lifetimes were in the range of days to weeks, which might or might not allow for efficient depollution depending on the river-water flow velocity.

## 1. Introduction

During the last three decades, the presence of emerging contaminants has been widely detected in surface and groundwaters around the world [1,2,3,4], posing a high environmental concern because of the adverse effects of these compounds on both aquatic organisms and human beings [5,6]. The so-called contaminants of emerging concern derive from anthropogenic activities; they include pharmaceuticals and personal-care products (PPCPs), pesticides, as well as nanoparticles. There are several ways by which the emerging contaminants (ECs) reach surface waters, including rivers, lakes, and the ocean, depending upon each compound and its use. For instance, PPCPs inputs to water bodies mainly derive from wastewater treatment plants (WWTPs), which are not able to completely degrade the received pollutant load [7]. In contrast, pesticides reach watercourses through soil runoff [8,9] and contamination of groundwater [10], which is an important water supply for rivers and lakes.

Although the persistence of both traditional pollutants and ECs is a huge problem for aquatic systems [11], sunlit water bodies can naturally trigger abiotic photochemical reactions that degrade dissolved compounds [12,13]. Indeed, pollutants can be degraded by reactive transients, called Photochemically Produced Reactive Intermediates (PPRIs), which are generated by the irradiation of chemical species that naturally occur in water bodies. The main PPRIs are hydroxyl and carbonate radicals (HO^•^ and CO_3_^•^^−^, respectively), singlet oxygen (^1^O_2_), and the excited triplet states of chromophoric dissolved organic matter (^3^CDOM*). CDOM is the main photosensitizer forming PPRIs in surface waters, followed by nitrate and nitrite. In particular, CDOM produces directly ^3^CDOM* and HO^•^ (although the exact pathway(s) to the latter species are not yet fully elucidated). Then, ^3^CDOM* react with dissolved oxygen (their main sink) to produce ^1^O_2_, and with CO_3_^2^^−^ to yield CO_3_^•^^−^. At the same time, HO^•^ radicals oxidize HCO_3_^−^/CO_3_^2^^−^ to CO_3_^•^^−^. Nitrate and nitrite directly produce HO^•^ radicals and then CO_3_^•^^−^ as a consequence of the reactions between HCO_3_^−^/CO_3_^2^^−^ and HO^•^ [14]. Water components can also inhibit the phototransformation processes. For instance, dissolved organic matter (DOM) scavenges both HO^•^ and CO_3_^•^^−^, and it can also inhibit the ^3^CDOM*-induced degradation of pollutants by back-reducing their partially oxidized reaction intermediates, thanks to its antioxidant moieties [15,16]. Besides indirect photochemistry through PPRI reactions, direct photolysis (i.e., direct transformation upon sunlight absorption) plays an important role in the whole degradation pathways of contaminants [17,18]. In general, all these reactions act as self-depollution processes for water bodies. However, they can sometimes induce the formation of compounds that are more harmful than the parent molecule [19,20]. Finally, it must be pointed out that biodegradation [21] and, to a lesser extent, chemical hydrolysis [22] are other efficient degradation processes for water pollutants [23].

Many works have been carried out to measure both the production of PPRIs, and their ability to degrade pollutants during lab experiments (see, e.g., in [24,25,26]). Unfortunately, the complexity of the aquatic environment does not easily allow for direct measurements in the field. As an alternative, by means of modeling tools one can calculate PPRIs formation and scavenging as well as pollutant photodegradation processes. The models take into account several environmental parameters; the exact reproduction of which would require a lot of efforts and time in laboratory experiments.

This modeling work deals with the assessment of the formation potential of the main PPRIs in European rivers, located in the mid-latitude belt between 40 and 50°N. The PPRIs steady-state concentrations were calculated and mapped over the relevant region, based on water chemistry data that are available in the GEMStat database [27]. This allowed us to assess how much these watercourses would be able to photodegrade water pollutants, such as isoproturon, carbamazepine, ibuprofen, atrazine, and clofibric acid, which have been widely detected in both surface and groundwater [28,29,30,31,32,33]. Some pollutants can even produce more toxic compounds by photodegradation, as in the case of carbamazepine and clofibric acid [34,35]. The approach to photochemical modeling used here has been previously validated, and shown to be able to predict the photochemical reactivity of the investigated compounds in surface-water bodies [36,37,38,39]. Indeed, for several aquatic environments we have found a good agreement between the modeled photochemical half-lives of ibuprofen, carbamazepine, atrazine, and clofibric acid, which are an indication of the persistence of these pollutants in a water body, and the lifetimes actually measured in the field (Table 1).

## 2. Results and Discussion

### 2.1. Mapping the PPRIs Steady-State Concentrations

The steady-state concentrations of ^3^CDOM* and HO^•^ radicals, referred to late spring/early summer fair-weather conditions, are here reported as average values for the period 1990–1995 (see the “Material and Methods” section for further details). The singlet oxygen (^1^O_2_) map is not shown here because it closely mirrors that of ^3^CDOM* [40]. The ^3^CDOM* map is reported in Figure 1, showing that ^3^CDOM* ranges between ~6.0 × 10^−17^ and ~9.0 × 10^−16^ mol L^−1^. These are reasonable concentration values for ^3^CDOM*, which are often measured during the irradiation of both synthetic and environmental DOM solutions [25,41]. Because both the water optical depth (0.1 m) and the DOM photochemical features were kept constant during modeling, the higher is the content of dissolved organic carbon (DOC, which is a measure of DOM, see Appendix A in the Appendix A), the higher is the steady-state concentration of ^3^CDOM*.

The steady-state concentrations of HO^•^ radicals are shown in Figure 2, and they deserve further explanation. Because (C)DOM is both the main scavenger, and (often) the main photochemical source of HO^•^ radicals, the DOC datum alone might theoretically be used to roughly assess the HO^•^ steady-state concentration ([HO^•^]). This modeling approach was here adopted as first approximation for those stations, for which the concentration values of the other HO^•^ sources and sinks (i.e., NO_3_^−^, NO_2_^−^, HCO_3_^−^, and CO_3_^2^^−^) were not available. Actually, because the role of NO_3_^−^ and NO_2_^−^ as HO^•^ sources is higher than the role of HCO_3_^−^, CO_3_^2^^−^, and NO_2_^−^ as HO^•^ scavengers [43,44,45,46], by using a DOC-only approach one gets a lower limit for the steady-state [HO^•^] (see Appendix A for a quantitative assessment of the differences, which are expected to occur in reasonable environmental scenarios).

Sufficiently complete datasets were available for France and Switzerland (see Appendix A for the NO_3_^−^ maps), for which consideration of all the main HO^•^ sources and sinks provided more realistic [HO^•^] levels. Therefore, the main reason for the difference in the calculated [HO^•^] between these two countries and the others is the availability of a more complete set of water chemistry data for the former. This issue needs consideration when the calculated lifetimes of pollutants are compared.

The data pertaining to six Swiss monitoring stations were characterized by the most complete dataset, which allowed for an extension of the investigated period over a dozen years (i.e., from 1990 to 2002), and for the calculation of the steady-state [CO_3_^•^^−^] as well. Figure 3a shows the time trends of the steady-state concentrations of HO^•^, ^3^CDOM*, and CO_3_^•^^−^ in Swiss rivers between 1990 and 2002. The reported results are values averaged among all the available watercourses, for the month of June of each year, and the associated standard deviations account for the geographical variability of the PPRIs concentrations.

As average values over all the Swiss stations, the steady-state [HO^•^] and [^3^CDOM*] did not undergo significant variations from 1990 to 2002. There were exceptions for the single stations, however, and particularly for Rekingen (Rhine river) and Port du Scex (Rhône river), which showed a statistically significant decrease of HO^•^ (Rekingen: r = −0.85, *p* = 9.2 × 10^−4^; Port du Scex: r = −0.66, *p* = 0.03) and an increase of ^3^CDOM* (Rekingen: r = 0.73, *p* = 0.01).

In the case of the steady-state [CO_3_^•−^], one observes an overall two-fold decrease of the values as all-station averages from 1990 to 2002 (Figure 3a). This substantial decrease would be the consequence of two parallel variations, namely a decrease of nitrate (indirect CO_3_^•−^ source via HO^•^ photogeneration) and an increase of the DOC (direct CO_3_^•−^ sink [46]).

The overall trend observed in Swiss rivers suggests an approximately constant behavior towards the photoinduced processes triggered by HO^•^ and ^3^CDOM*, but an important loss in the ability to carry out photodegradation by CO_3_^•−^. The latter issue can have an impact over the photochemical lifetimes of the compounds that are mainly (or otherwise considerably) degraded by CO_3_^•−^, such as electron-rich phenols, anilines, and sulfur-containing molecules [47].

### 2.2. Modeling the Photodegradation of Contaminants

The Europe-wide maps reported in Figure 1 and Figure 2 can be used to get insight into the photodegradation kinetics of contaminants that mainly react with HO^•^ and ^3^CDOM*. However, just because of data-set differences, the [HO^•^] map provides higher values for France and Switzerland while underestimating the levels in the other investigated European countries. To minimize this bias, one should consider a model compound that mainly undergoes degradation by ^3^CDOM*. Phenylurea herbicides belong to this category [48] and, in particular, isoproturon (IPT) is a compound that has a second-order reaction rate constant with ^3^CDOM* that is not so far from that with HO^•^ ($k_{IPT+HO^{•}}$ = 7.9 × 10^9^ L mol^−1^ s^−1^, $k_{IPT+{}^{3}CDOM*}$ = 3.2 × 10^9^ L mol^−1^ s^−1^), while also undergoing limited directphotolysis [49,50]. IPT is typically used in cereals and undergoes runoff processes when applied to the crops [51,52,53], making it a potential water contaminant. For this reason, IPT has been detected in several rivers during the 1990s [54,55,56], and it is often detected around Europe nowadays as well [57,58,59]. IPT is moderately toxic toward aquatic organisms, showing both additive and synergistic toxic effects with other pesticides [60,61].

The European map of IPT half-life times is reported in Figure 4. It appears that IPT photodegradation would be faster in Central Europe, with photochemical half-lives of less than one month, because of higher [^3^CDOM*] values. In particular, in France and Switzerland, the half-life time would be in the range of a few days to two weeks. The case of the rivers located in SW France is quite interesting, because relatively fast IPT degradation is foreseen there despite the low [^3^CDOM*] values shown in Figure 1. The reason is that the same locations showed very high [HO^•^] levels (see Figure 2), which could offset the limited role of ^3^CDOM*. Elevated HO^•^ values would be found in those stations because of the combination of moderately high nitrate concentration with quite low DOC, which enhances HO^•^ photogeneration while minimizing its scavenging.

It is also interesting to assess the photodegradation kinetics of different pollutants over a longer time interval, to see to what extent variations in photochemistry can affect the self-depollution ability of rivers. As already mentioned in Section 2.1., this assessment was possible in the case of Swiss rivers for the period 1990–2002. Moreover, the Swiss dataset was complete enough to allow for the calculation of the degradation kinetics of clofibric acid (hereinafter CLO), ibuprofen (IBP), atrazine (ATZ), and carbamazepine (CBZ). The main involved photodegradation pathways are HO^•^ and ^3^CDOM* (CLO), HO^•^ and direct photolysis (IBP), mostly HO^•^ (CBZ), as well as HO^•^, CO_3_^•−^ and ^3^CDOM* with comparable weight (ATZ) (see Appendix A).

On average over all the Swiss stations, the photodegradation kinetics of CLO, IBP, and CBZ would not change much (see Figure 5). This result reflects the limited overall variability in the steady-state [HO^•^] and [^3^CDOM*] that was reported in Figure 3a. The scenario is slightly different in the case of ATZ, with somewhat longer half-lives in the period 1994–2002 compared to 1990–1993. This finding is a consequence of the fact that the ATZ half-life showed a statistically significantly increase over time in five out of the six stations, the only exception being Chancy (located along the Rhone river, see Figure 3b). Actually, among the considered contaminants, ATZ is the only one undergoing important photodegradation by reaction with CO_3_^•−^, and this transient species is expected to decline over the investigated time period (see Figure 3a).

Even for those contaminants, for which the overall photodegradation kinetics did not change much over time as an all-station average, there were some important differences as far as the single sites are concerned. As already mentioned in Section 2.1., the calculated steady-state [HO^•^] had a significant decline in the Swiss stations of Rekingen and Port du Scex. This trend is reflected into a statistically significant increase of the lifetimes of CLO (Rekingen: r = 0.83, *p* = 4.1 × 10^−4^; Port du Scex: r = 0.67, *p* = 0.01), IBP (Rekingen: r = 0.81, *p* = 7.3 × 10^−4^; Port du Scex: r = 0.67, *p* = 0.01) and CBZ (Rekingen: r = 0.83, *p* = 3.9 × 10^−4^; Port du Scex: r = 0.68, *p* = 0.01). In the case of CBZ, a statistically significant lifetime increase was also assessed in the station of Brugg (r = 0.66, *p* = 0.03), because of the parallel decrease of both [HO^•^] and [CO_3_^•−^].

In the case of CBZ, we were also able to calculate the formation yield of acridine, which is a mutagenic transformation intermediate formed from CBZ upon direct photolysis and HO^•^ reaction [34,37]. The acridine formation yield was calculated as the ratio between the formation rate constant of acridine and the transformation rate constant of CBZ [62]. The acridine yield did not change much over time, and it was always ~2.5% with limited variability among the different stations (Appendix A). Although not exceedingly high, this formation yield of a mutagenic compound would pose some concern. As mentioned before, the photodegradation kinetics of CBZ would slow down over time in some stations and, at constant yield, the formation rate constant of acridine is predicted to undergo a parallel decrease. In those cases, the loss in performance towards self-depuration would be offset by a correspondingly lower potential to produce a harmful transformation intermediate.

The lifetimes reported in Figure 5 show some long-term trends and reflect the variability among stations, but they range overall between 5 and 25 days. The lowest photochemical persistence is predicted for IBP, and the highest for ATZ. When considering a water flow velocity of 1 m s^−1^ [63,64], these lifetimes would translate into river half-life lengths ranging between around 400 and over 2000 km. These are the river lengths that would be required for the pollutants to halve their concentration due to photochemical depollution [65], and they mean that photodegradation would be hardly complete when river water leaves Switzerland. The scenario would become quite different under water-scarcity conditions, when the water flow velocity can decrease down to 0.1 m s^−1^ [66]. In this case, with half-life river lengths ranging between 40 and 200 km, river water photodegradation of contaminants would become much more important. This finding confirms earlier assumptions, according to which photochemistry could be an important self-depollution process in rivers under drought conditions, in which case photodegradation may help cope with the increased pollution caused by lower dilution of, e.g., wastewater discharges [65].

## 3. Materials and Methods

The investigation area comprised European rivers located in the latitude belt between 40 and 50°N (Appendix A). The PPRIs steady-state concentrations were calculated for these water bodies in the month of June, when the highest photoactivity of surface waters should occur. For calculations, we used photochemistry-modeling software derived from the Aqueous Photochemistry of Environmentally-occurring Xenobiotics (APEX) code [62]. Although the software version we used here is not yet available for public use, it has already been employed by us for similar modeling purposes in recently published work [67].

Solar spectral irradiance, optical water depth, and photosensitizer concentrations were the main input data for computation. The sun spectrum reaching the Earth surface in June at 45°N during clear-sky conditions was used [68], as it is reasonably representative of the considered latitude belt. Despite some differences between the spectral irradiance at 40°N and at 50°N (Appendix A), variations above 310 nm are within the uncertainty of the photochemical model (<20%) and the spectra can be considered very similar. Modeling was performed with 0.1 m as the water optical depth (*d*), which allowed for a normalization of the computations without the need of taking into account unavailable seasonal hydrology variations in rivers, under different weather conditions.

Concentration data of the relevant photosensitizers (CDOM, nitrate, and nitrite) were obtained from the Global Environment Monitoring System website [27]. This website reports a wide dataset of water quality parameters around the globe. Suitable data allowing for photochemical modeling were practically available for Europe only, and they date back to the 1990s. We used the dissolved organic carbon (DOC) to quantify both DOM and CDOM, the latter assessed as the water absorption spectrum, on the basis of literature-reported correlations for inland surface waters [62]. The DOC data were not available for all the monitoring stations, because some stations reported instead the chemical oxygen demand (COD). Fortunately, in the case of surface waters the values of DOC and COD are correlated. To have a reasonably wide investigated area/period, we used the equation *DOC = 0.18∙COD* to assess the DOC when it was not available [69]. This correlation has already been used in previous work to model the photochemistry of Lake Peipsi (Estonia) [70]. The DOC maps are reported in the Appendix A (Appendix A). With the DOC datum it was possible to compute the PPRIs formation rates from sunlit CDOM, as well as their scavenging by DOM. The main assumption here was to consider a uniform, average behavior of (C)DOM as both PPRIs photosensitizer and scavenger. Indeed, although the (C)DOM type depends upon its source and the transformation processes it has undergone [71], variation of the (C)DOM photochemical properties could be less important than the natural environmental variability might suggest [72]. When available, nitrate (Appendix A) and nitrite concentrations data were used to compute HO^•^ and CO_3_^•−^ radical photochemistry. The calculation of the CO_3_^•−^ steady-state concentration also required knowledge of the levels of HCO_3_^−^ and CO_3_^2−^, because these anions are the main precursors of CO_3_^•−^ when oxidized by HO^•^ and ^3^CDOM* [45]. In particular, alkalinity and pH data from GEMStat were used to calculate [HCO_3_^−^] and [CO_3_^2−^], with an iterative computation procedure explained in detail elsewhere [67]. Finally, we used input data included in the period between 1990 and 1995, where the dataset was most complete throughout Europe. In general, only those stations with DOC data for at least four years in this period were considered for modeling.

The photodegradation of isoproturon (IPT), clofibric acid (CLO), ibuprofen (IBP), atrazine (ATZ), and carbamazepine (CBZ) was assessed by calculating their half-life times (*t*_1/2_) as *t*_1/2_
*= ln2*(*k’*)^−1^. Here, *k’* (Equation (1)) is the pseudo-first order degradation rate constant of the considered contaminant C, accounted for by both direct photolysis and indirect photochemistry:(1)$$k\prime =\sum_{i}\;([i]\cdot \;k_{i+C})+\Phi_{C}\sum_{j}\left[\Delta \lambda_{j}\;p_{\lambda_{j}}^{o}\;A_{\lambda_{j},C}(1-10^{-A_{\lambda_{j},CDOM}}\right)\;{\left(A_{\lambda_{j},CDOM}\right)}^{-1}]$$

In Equation (1), $k_{i+C}$ is the second-order rate constant for the reaction between the contaminant and the transient (PPRI) *i* (*i* = HO^•^, CO_3_^•−^, ^3^CDOM*); $\Phi_{C}$ is the direct photolysis quantum yield of the contaminant (see Table 2 for the relevant values); [*i*] is the steady-state concentration of the PPRI; and $A_{\lambda_{j},C}$ and $A_{\lambda_{j},CDOM}$ are the spectral absorbances of, respectively, the contaminant and CDOM at the relevant wavelengths. The light-absorption calculations were based on the Bouguer–Beer–Lambert Law. In particular, $A_{\lambda_{j},CDOM}$ was computed with the relationship $A_{\lambda_{j},CDOM}=0.45\;d\;DOC\;e^{-0.015\;\lambda_{j}}$, which gives a suitable absorption spectrum for surface freshwaters [62]. Finally, $p_{\lambda_{j}}^{o}$ is the average solar spectral photon flux density reaching the Earth surface at 45°N during a fair-weather day in June. The sum in Equation (1) was calculated for increments $\Delta \lambda_{j}$ = 1 nm.

In the case of CLO, the rate constant for the reaction with ^3^CDOM* took into account the back-reduction effect due to the antioxidant moieties of DOM. We used the correction factor *φ =* (1 − *f*) *+ f*∙[1 + *DOC* × (*DOC*_1/2_)^−1^]^−1^ (see, e.g., in [15,16]). *DOC*_1/2_ is a parameter that quantifies the back-reduction extent (the lower is *DOC*_1/2_, the more important is the back reduction), whereas *f* is the fraction of CLO that undergoes back-reduction when reacting with ^3^CDOM*. The relevant values are *DOC*_1/2_ = 0.1 mg_C_ L*^−^*^1^ and *f* = 0.32 [73].

CBZ yields mutagenic acridine by both direct photolysis and reaction with HO^•^. In particular, the acridine yields from CBZ in the two processes are, respectively, $\eta_{d.p.}$ = 0.036 and $\eta_{HO^{•}}$ = 0.031, whereas no acridine is formed upon reaction of CBZ with CO_3_^•^*^−^* or ^3^CDOM* [37]. The overall yield of acridine from CBZ was calculated as follows [62],
(2)$$\eta_{acridine}=\frac{\eta_{d.p.}{\left({k^{\prime }}_{CBZ}\right)}_{d.p.}+\eta_{HO^{•}}{\left({k^{\prime }}_{CBZ}\right)}_{HO^{•}}}{{\left({k^{\prime }}_{CBZ}\right)}_{tot}}$$
where ${\left({k^{\prime }}_{CBZ}\right)}_{d.p.}$ is the first-order degradation rate constant of CBZ accounted for by direct photolysis, ${\left({k^{\prime }}_{CBZ}\right)}_{HO^{•}}$ the first-order CBZ rate constant for degradation by HO^•^, and ${\left({k^{\prime }}_{CBZ}\right)}_{tot}$ the overall rate constant of CBZ photodegradation accounted for by all the photoinduced processes (direct photolysis, HO^•^, ^3^CDOM*, and CO_3_^•^^−^).

## 4. Conclusions

Based on data of water chemistry, we were able to produce a series of maps that describe the photochemistry of mid-latitude European rivers during the first half of the 1990s. In particular, the available data allowed us to map the steady-state [^3^CDOM*] and [HO^•^] as well as [^1^O_2_], which is very similar to [^3^CDOM*]. The maps thus obtained allowed us to model the photodegradation kinetics of the phenylurea herbicide isoproturon, which is predicted to occur faster in Central Europe. In several French rivers, the isoproturon lifetime would be shorter than a week.

The most complete dataset was available for the Swiss rivers, which allowed us to extend the investigation to a longer time period (1990–2002). In that case, we observed limited variations in the photochemistry of HO^•^ and ^3^CDOM*, in terms of both their steady-state concentrations and the ability to degrade water pollutants. However, CO_3_^•^^−^ radicals underwent a substantial decrease, which would for instance result in slower photodegradation of atrazine, as well as of other contaminants that undergo fast reaction with CO_3_^•^^−^ (electron-rich phenols, amines, and sulfur-containing species).

Dataset incompleteness prevented us from extending the model to a more recent time period. For an updated European photochemical map to be produced, and possibly compared with the present data to highlight the main variations over the years, there would be the need to set-up a geographically wide database of recent water-chemistry data. They should include at least DOC, nitrate, nitrite, alkalinity, and pH. In this way, it would be possible to get a more comprehensive view of how surface waters can naturally cope with human-induced pollution, also highlighting the most vulnerable environments from this respect.

## Figures and Tables

**Figure 1 molecules-25-00424-f001:**
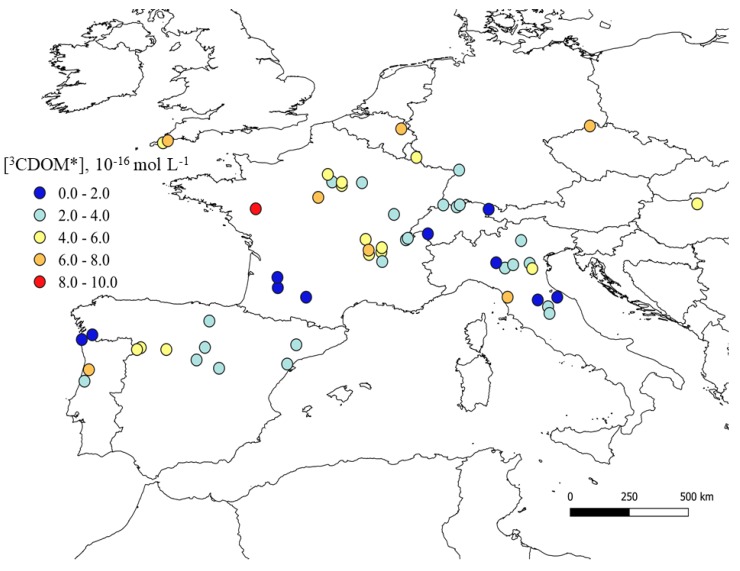
^3^CDOM* map of European rivers located in the latitude belt from 40 to 50° N. Data refer to the month of June and are average values over the period 1990 to 1995. The map was made by means of the QGIS software (version 3.2.2 ‘Bonn’ [42]).

**Figure 2 molecules-25-00424-f002:**
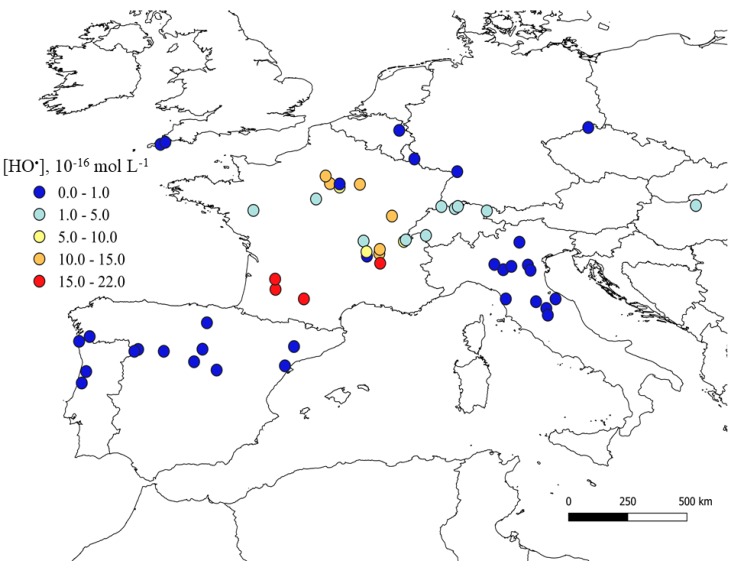
HO^•^ radicals maps of European rivers located in the latitude belt from 40 to 50° N. Data refer to the month of June and are average values over the period 1990–1995. The maps were made by means of the QGIS software (version 3.2.2 ‘Bonn’ [42]). Note that, apart from France and Switzerland, the other calculated [HO^•^] values should be considered as lower limits.

**Figure 3 molecules-25-00424-f003:**
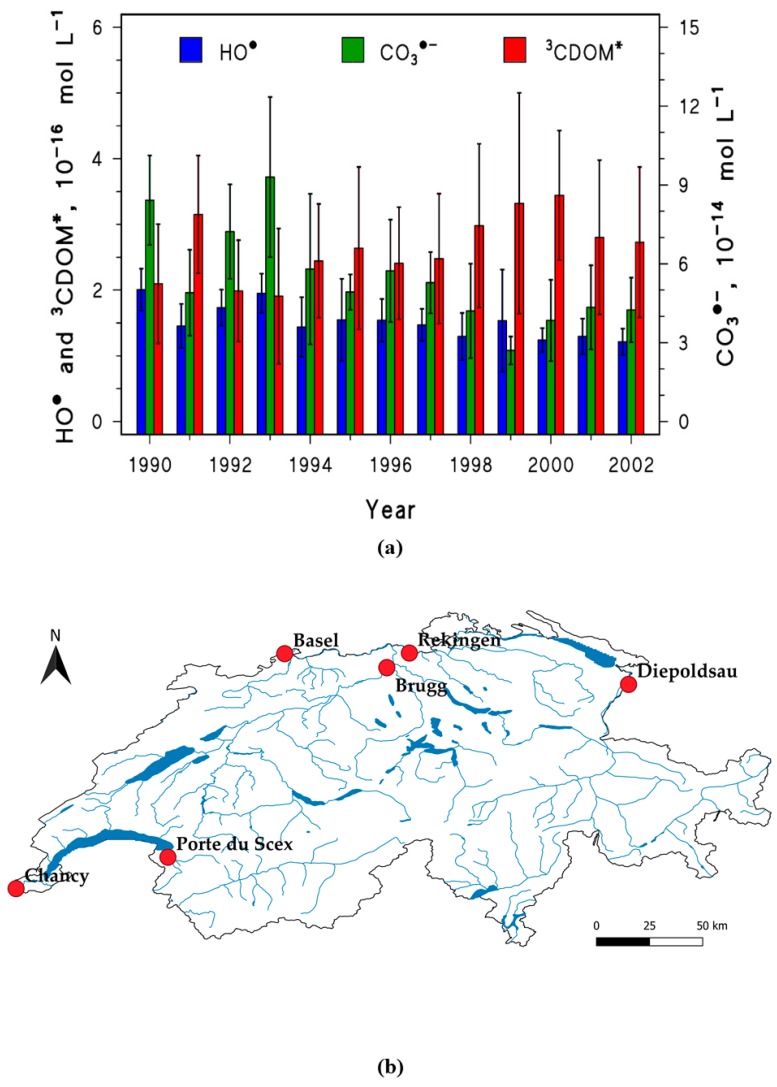
(**a**) PPRIs steady-state concentration trends for Swiss rivers during the month of June 1990–2002. The relevant input data for modeling the PPRIs concentrations were (C)DOM, NO_3_^−^, HCO_3_^−^ and CO_3_^2^^−^, plus 0.1 m as the water optical depth. Error bars are standard deviations accounting for the spatial variability among the different stations, shown in panel (**b**).

**Figure 4 molecules-25-00424-f004:**
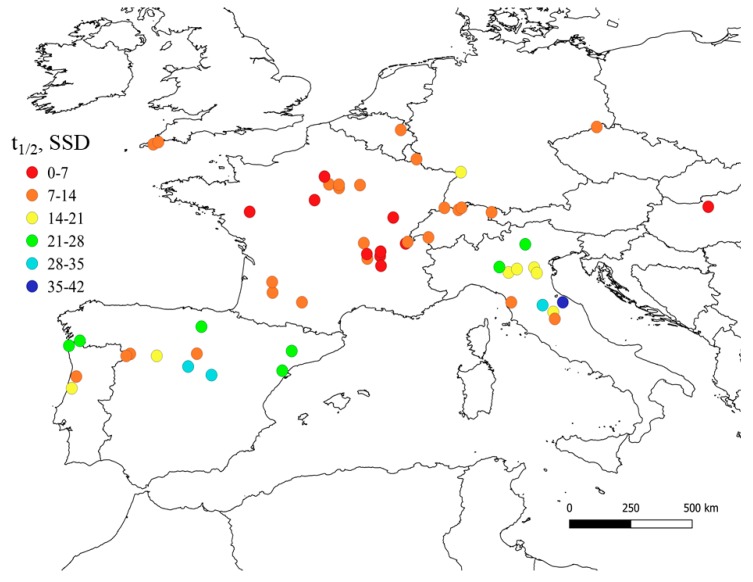
Map of IPT photochemical half-life times (t_1/2_) for European rivers located in the latitude belt from 40 to 50° N. Data refer to the month of June (SSD = sunny summer days) and are average values over the period 1990–1995. The relevant input data for modeling t_1/2_ were from Figure 1 (^3^CDOM*) and Figure 2 (HO^•^ radicals), and the lifetime was calculated as $t_{1/2}=\ln2\;{\left(k_{IPT+HO^{•}}[HO^{•}]+k_{IPT+{}^{3}CDOM*}[{}^{3}CDOM*]\right)}^{-1}$. The maps were made by means of the QGIS software (version 3.2.2 ‘Bonn’; [42]).

**Figure 5 molecules-25-00424-f005:**
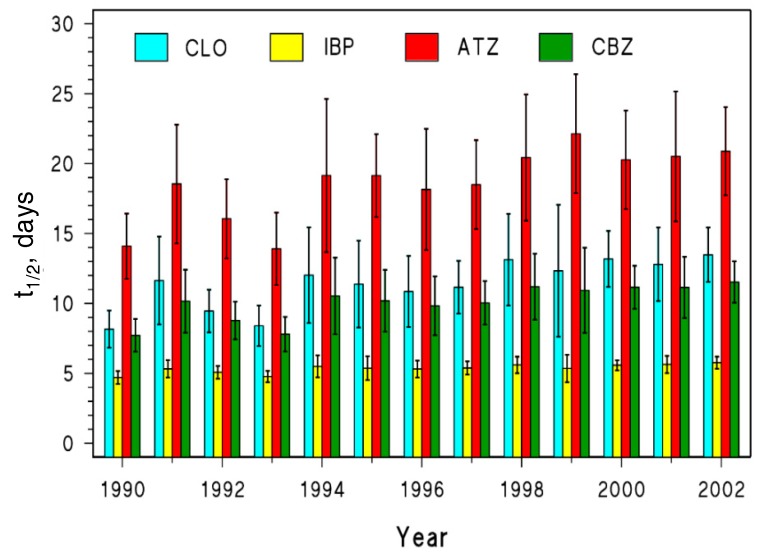
Photochemical half-life time (t_1/2_) trends for CLO, IBP, ATZ and CBZ in Swiss rivers (0.1 m as optical water depth) during the month of June for 1990–2002. Error bars are standard deviations accounting for the spatial variability of the model results.

**Table 1 molecules-25-00424-t001:** Literature comparisons between the modeled and the field-measured photochemical half-life times of some of the contaminants investigated in the present work.

Compound	Photochemical Half-Life Time, Days		Location	Ref.
	Model	Field		
Ibuprofen	60 ± 10	60–115	Lake Greifensee	[36]
Carbamazepine	110 ± 45	140 ± 50	Lake Greifensee	[37]
Atrazine	17 ± 4	20–21	Chesapeake Bay (1 m depth)	[38]
	64 ± 18	67–100	Chesapeake Bay (10 m depth)	
Clofibric acid	60–120	70	Lake Greifensee	[39]

**Table 2 molecules-25-00424-t002:** Photoreactivity parameters used to assess the photochemical half-life times of the relevant contaminants. Note that the investigated compounds undergo negligible degradation by ^1^O_2_.

Contaminant	(Units of L mol^−1^ s^−1^)		(Unitless)	
	$k_{HO^{•}+C}$	$k_{CO_{3}^{•-}+C}$	$k_{{}^{3}CDOM*+C}$	Φ_C_
CLO	1.2 × 10^10^ [39]	Negligible [39]	φ∙3.6 × 10^9^ [73]	5.5 × 10^−3^ [39]
IBP	1.2 × 10^10^ [36]	1.2 × 10^6^ [74]	1.5 × 10^9^ [73]	0.33 [36]
ATZ	3.0 × 10^9^ [75]	6.2 × 10^6^ [76]	7.2 × 10^8^ [73]	1.6 × 10^−2^ [38]
CBZ	9.0 × 10^9^ [77,78,79]	4.2 × 10^6^ [74]	7.6 × 10^8^ [37]	7.8 × 10^−4^ [37]

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
