# Peer review of "Mapping the Photochemistry of European Mid-Latitudes Rivers: An Assessment of Their Ability to Photodegrade Contaminants"

_molecules, 2020, doi:10.3390/molecules25020424_

Round 1

Reviewer 1 Report

With the increasing adverse effects of photochemically produced reactive intermediates (PPRIs) resulting from the anthropogenic activities, this study appears to be an advance in the prediction and assessment environmental toxic contaminants. In particular, the idea of modeling work to track the PPRIs clearly of European rivers, located in the mid-latitude belt between 40 and 50°N allow to monitor how much these watercourses would be able to photodegrade water pollutants. Therefore, this manuscript is enough to publish in molecules journal after minor revision. One thing that I recommend during the revision process is that describing the previous validation reports somewhere in discussion if the authors feel it is relevant.

Author Response

We agree with reviewer #1, and also reviewer #2 gave a similar recommendation. To make things clearer, we have reported Table 1 at the end of the introduction where a comparison between field data and model predictions are reported, based on previously published works.

Reviewer 2 Report

In this submitted work, the authors used modeling method to deal with the assessment of the production of photochemically produced reactive intermediates (PPRIs) in European rivers, which were able to degrade organic pollutants. Although the conditions in natural rivers are much more complicated, this study is meaningful and interesting. The main concern is that has this modeling method been proven to be suitable for estimating the generation of PPRIs and the degradation of pollutants. If done, I recommend it to be published.

Author Response

We agree with reviewer #2, and also reviewer #1 gave a similar recommendation. To make things clearer, we have reported Table 1 at the end of the introduction where a comparison between field data and model predictions are reported, based on previously published works.